# Network Lifetime Improvement through Energy-Efficient Hybrid Routing Protocol for IoT Applications

**DOI:** 10.3390/s21227439

**Published:** 2021-11-09

**Authors:** Mukesh Mishra, Gourab Sen Gupta, Xiang Gui

**Affiliations:** Department of Mechanical and Electrical Engineering, School of Food and Advanced Technology, Massey University, Palmerston North 4442, New Zealand; G.SenGupta@massey.ac.nz (G.S.G.); X.Gui@massey.ac.nz (X.G.)

**Keywords:** energy efficiency, network lifetime, hybrid routing protocol, genetic algorithm, particle swarm optimization

## Abstract

The application of the Internet of Things (IoT) in wireless sensor networks (WSNs) poses serious challenges in preserving network longevity since the IoT necessitates a considerable amount of energy usage for sensing, processing, and data communication. As a result, there are several conventional algorithms that aim to enhance the performance of WSN networks by incorporating various optimization strategies. These algorithms primarily focus on the network layer by developing routing protocols to perform reliable communication in an energy-efficient manner, thus leading to an enhanced network life. For increasing the network lifetime in WSNs, clustering has been widely accepted as an important method that groups sensor nodes (SNs) into clusters. Additionally, numerous researchers have been focusing on devising various methods to increase the network lifetime. The prime factor that helps to maximize the network lifetime is the minimization of energy consumption. The authors of this paper propose a multi-objective optimization approach. It selects the optimal route for transmitting packets from source to sink or the base station (BS). The proposed model employs a two-step approach. The first step employs a trust model to select the cluster heads (CHs) that manage the data communication between the BS and nodes in the cluster. Further, a novel hybrid algorithm, combining a particle swarm optimization (PSO) algorithm and a genetic algorithm (GA), is proposed to determine the routes for data transmission. To validate the efficacy of the proposed hybrid algorithm, named PSOGA, simulations were conducted and the results were compared with the existing LEACH method and PSO, with a random route selection for five different cases. The obtained results establish the efficiency of the proposed approach, as it outperforms existing methods with increased energy efficiency, increased network throughput, high packet delivery rate, and high residual energy throughout the entire iterations.

## 1. Introduction

A wireless sensor network (WSN) is a collection of small sensing devices (nodes) that performs communication with other devices through a wireless channel [1]. The prime characteristics of SNs in WSNs are low cost, small size, low computational power, communication within short distances, and multifunctional capabilities such as sensing, routing, and data processing, [2]. The processing capabilities of SNs include sensing data from the environment and communicating the collected data to the BS [3]. This transmission of data among the SNs and the BS requires energy to be expended. Often, the energy consumed is more than the actual energy requirement, as there may be wastage of energy due to various factors. An example of a factor that causes energy wastage is the transmission of redundant data [4]. Further, the transmission of data between the SNs and the BS may increase demand in cases of larger geographical areas owing to the hostile nature of the environment [5,6]. Energy consumption may also vary significantly between single- and multi-hop communication [7]. To address these issues, hierarchical routing and the clustering of SNs have demonstrated proven competence to enhance the lifetime of networks. [8]. The selection of CHs further reduces energy consumption, as it collects data from the cluster members (CMs) and forwards the same to the BS through the CH [9]. In general, most modern IoT devices perform faster data collection and, hence, require faster data processing and transmission to the BS [10]. 

The routing protocol aims to select the optimal path for data transmission, which is a challenging task. The selection of an optimal path highly depends on the various network parameters, such as channel characteristics, network type, and performance metrics [2]. In smaller IoT networks, the BS and SNs are in closer proximity and, therefore, communication may take place directly in a single hop. In contrast, the communication in large-scale IoT networks uses multi-hops, as direct communication with the BS may not be feasible. This can be attributed to many reasons—radio power, bandwidth, energy, or memory [3]. Hierarchical routing algorithms aim to enhance the network throughput and lifetime of WSNs in various geographical deployments. However, they may achieve energy efficiency only to a limited extent, which motivated the authors to present a cluster-based optimization, centralizing energy efficiency [5], scalability [6], complexity [7], and robustness [8]. Additionally, the efficacy of hierarchical clustering approaches can be further enhanced using a particle swarm optimization–genetic algorithm, a multi-objective optimization model.

The optimization of a network’s lifetime is a challenging and vital issue in WSNs and, thus, has grabbed the attention of various researchers, as it is required in order to conserve energy [11,12]. According to some researchers, the network lifetime can be considered the time elapsed until the first sensor node loses all its energy in the network. In order to optimize the network lifetime, researchers have been working in the direction of optimizing various parameters, namely hop count, path reliability, energy consumption, and so forth. The authors of this paper attempted to improve the network lifetime through the optimization of the routing protocol, hop count, and reliable path. In the proposed approach, the authors used factors such as residual energy, hop count, and the reliable path to the sink in order to maximize the network lifetime. The performance of the proposed approach was validated with various metrics, such as packet delivery, throughput, energy consumption, and more.

The following are the main contributions of the current research:The integration of the GA and PSO algorithms for optimal routing to transfer data from the CHs to the BS. The authors of this paper propose a PSOGA-based routing protocol that works towards the optimization of the network lifetime of WSNs. The proposed protocol determines a score based on the residual energy, buffer, hop count, and reliable path. The evaluated score is used to determine the next hop node in the path.The authors redefined network lifetime using three important aspects of a network, namely the number of alive nodes, sink connectivity, and successful packet delivery. Further, a performance evaluation framework was employed based on the redefined network lifetime to evaluate the performance of routing protocols.The proposed approach was implemented and tested for different scenarios with varying grid sizes and node densities.The competence of the proposed approach was established by deploying the BS at the edges of the grid.

The paper is structured as follows: Section 2 presents related works; Section 3 discusses the proposed PSOGA model; Section 4 discusses the proposed hybrid PSOGA model; Section 5 presents the network setup; Section 6 presents a comparison between the results of the proposed PSOGA model with the PSO and LEACH protocols; Section 7 concludes the paper with suggestions for possible avenues of future research.

## 2. Related Work

Numerous authors have undertaken research with the aim of devising effective routing for WSNs. This section presents some of the prominent results and findings by such researchers in the field.

Anees, J., et al. [13] suggested a delay aware and energy-efficient opportunistic node selection in restricted routing (DA-EEORR). The authors claim the suggested model to be novel and suitable for a delay-sensitive environment. The proposed model attains a promising balance between energy consumption and average end-to-end delay by finding an optimal path. The model uses the idea of an opportunistic connection random graph (OCRG) to select the next hop. OCRG is further used to calculate optimal path connectivity, using factors such as transmission frequencies, residual energy, link quality, and so forth. The concept of a restricted research space was also employed in the proposed model to find the minimum distance next-hop node. The simulation findings advocate that the proposed method outperforms the current standards. This outperformance of the suggested approach can be witnessed in terms of the network lifespan, power usage, the overhead of the control packet, and the packet delivery ratio. Most of the findings showed only a marginal improvement since the study was focused more on path correction and the tracking of routes rather than establishing the optimal path in a hierarchical network. Thus, the proposed work achieves great balance between energy consumption and end-to-end delay; however, the work can still be extended further in the direction of incorporating multiple sink nodes to realize realistic delay-sensitive applications.

The authors Ullah, F., et al. [14] presented research on optimal route selection in wireless body area networks (WBANs). A WBAN is a network of miniaturized wearable sensing and computing devices that communicate the sensed data around a human body, and hence, they have been excessively used in remote patient monitoring, sports activity monitoring, and so on. They may be used to monitor vital physical parameters, such as electro-cardiograph (ECG), electro-encephalography (EEG), and so on. Now, as WBANs are resource constrained, they necessitate efficient and energy-efficient routing mechanisms. The authors of [14] proposed an energy-efficient and reliable routing scheme (ERRS) to increase reliability and resource stability in WBANs. To achieve this, the suggested approach implements two solutions, namely the selection of the forwarder node and the rotation of the forwarder node. EERS employs adaptive static clustering routing to achieve an enhanced stability period and a prolonged network life. During the simulation of the proposed approach, it was observed that the EERS achieved an improvement of 26% over the established protocols. The performance of the EERS is measured in terms of throughput and network stability. Additionally, it achieved an improvement of 17% and 40% in terms of end-to-end delay over the SIMPLE and M-ATTEMPT protocols, respectively, establishing the supremacy of the proposed algorithm. Although the study provided an efficient solution for WBANs during the simulation, motivating the researchers, its scalability and mobility remain challenges that need to be addressed in order for its real-life and widespread deployment.

The need for longer network lifetimes and fast data transmissions for unattended time-sensitive nodes was also recognized by Maurya, S., et al. [15]. In their study, the authors claimed that most of the routing approaches for such applications rarely consider all related issues at once, such as network traffic, the loss of packets, and energy consumption. Another associated challenge mentioned by the authors is the consideration of a homogeneous sensor network, as real-life deployments need to handle heterogeneous nodes. To address these challenges, the authors of [15] proposed a novel delay-aware energy-efficient reliable routing (DA-EERR) technique that considers heterogeneous nodes. The proposed approach defines a restricted search space to ensure the timely delivery of time-sensitive data. Thereafter, an algorithm selects a delay-aware energy-balanced path between the source and the sink within the search space to ensure fast communication. The suggested approach attains an improvement in successfully receiving data packets at the sink in large networks. The proposed routing method achieves significant improvement over comparative models for large and densely deployed networks. However, for smaller and sparse networks, the proposed approach will introduce a control packet overhead that may cause the quick cessation of the ring node, thus limiting the application of the proposed approach to large networks only.

The authors of [9] addressed the obstruction of energy constraints in wireless sensor networks as a motivating factor for their rigorous research to develop energy-efficient routing protocols. The authors attempted to propose a new protocol, namely the equalized CH election routing protocol (ECHERP), which conserves energy using balanced clustering. The proposed model uses the Gaussian elimination algorithm to evaluate the node combination to select the CH. The comparative evaluation of the proposed model establishes its effectiveness over standard protocols in terms of energy efficiency. Similarly, the authors of [16] also employed an energy-efficient dynamic clustering technique to organize nodes in WSNs. Their model estimates the count of active nodes using signals received from neighboring nodes. In addition, it computes the probability of an active node becoming a CH based on the energy requirement for inter-cluster and intra-cluster communication. The objective of the model is to maximize the network life. A simulation of the proposed approach demonstrated that the clustering method has the ability to scale well for large-scale WSNs also. The authors aimed to extend the work by evaluating the application feasibility of the proposed technique to general WSNs.

From the above studies done by various researchers, it is evident that significant research is being conducted on energy-efficient routing in various network models. Among the various approaches, clustering has demonstrated its competence in achieving energy efficiency and, thus, has motivated the authors to pursue research in this direction.

## 3. Network Energy Models (Methods)

This section presents the models employed in this study. Initially, the system model is presented, along with the necessary assumptions that have been made. Further, the energy consumption model based on the first order radio model for transmission and reception is presented. Subsequently, the proposed network model is also presented.

### 3.1. System Models and Assumptions 

We considered a two-dimensional network model [17,18] with sensor nodes, as well as the following assumptions:All sensor nodes are considered stationary.In this study, we assumed there was one BS for which the data collected from the source IoT nodes are destined.The homogeneous SNs had similar processing and communication capabilities. In addition, we considered that the SNs are deployed with the same initial energy.The SNs deployed randomly are always located with their x and y coordinates in the topological area.The distance between the two neighboring SNs was evaluated using Euclidean distance.

### 3.2. Energy Consumption Models

We used a first order radio model as the energy consumption model for the purpose of the transmission/reception of messages of the same length, *n*-bits. This model computes energy consumption $E_{i\;}$ for the transmission of IoT nodes during one round. The energy consumed to transmit *n*-bits of data to a node located at a distance *d* (using Euclidean distance metric) was estimated as follows [9]:(1)$$E_{tx}\left(d\right)=\left(\frac{n}{r}\right)\left(\Phi_{amp}d^{\alpha }+\Phi_{cir}\right)\;|\;\alpha =\{\begin{array}{c} 2,\;d<d_{cr}\\ 4,\;d>d_{cr} \end{array}\;\;$$
where $\Phi_{cir}$ represents the power consumption during the operation of the transmitter circuit, and $\Phi_{amp}$ is the power consumed by amplifier. *α* is the exponent indicating the path loss component with the range [2, 4], and $d_{cr}$ is the crossover distance based on free path loss and multipath loss. Here, *α* is 4 and 2 for multi-path loss and free path loss, respectively. The consumption of energy at the receiving node at rate *r* depends entirely on the operation of the circuit, which is represented as follows:(2)$$E_{rx}\left(d\right)=(n_{\;}/r_{\;})\Phi_{cir}\;$$

Thus, for an intermediate sensor node *i* at a single-hop distance, the energy consumption *E_i_* for transmission and reception for relaying over the distance *d* is given as follows:(3)$$E_{i}\left(d\right)=E_{tx}\left(d\right)+E_{rx}\left(d\right)=\left(\;{\frac{n}{r}}_{\;}\right)\left(\Phi_{amp}d^{\alpha }+2\Phi_{cir}\right)$$

### 3.3. Proposed Network Model

The details of the proposed model are given as follows:Step 1: All the SNs are static in the network.Step 2: The initial step is to define the network dimensions; the proposed model will have A × B dimensions.Step 3: The network area will be split into multiple grids or block clusters. In our model, it can be 2 × 4 = 8 grid clusters (comprising 100 nodes), 4 × 4 = 16 grids clusters (comprising 100 and 200 nodes) and 10 × 10 = 100 grids clusters (comprising 625 and 1250 nodes).Step 4: A CH for each grid is selected using the concept of trust model [19].Step 5: All the SNs in a particular grid sense the environment and transmit the sensed data to the CH.Step 6: The path between the CHs towards the sink is generated by considering the paths between adjacent grids in a zigzag fashion, which means that the area of interest is divided into numerous zigzag patterns so that each pattern has line segments and corners and each node is deployed in the corner of the zigzag pattern. The zigzag pattern enables a high coverage efficiency of 91%. In addition, it helps to cover the whole area of interest using the minimum number of nodes, thus generating the minimum coverage redundancy [20]. The score of each route is calculated using Equation (4) [21], where the weight factors are designated to assign the weights to the subcomponents and are configured as *α* + *β* + *γ* + *δ* = 1.
(4)$$Score\left(P\right)=\omega \alpha +min.buffer\left(p\right)\beta +\left(\frac{1+maxHC-HC\left(P\right)}{maxHC}\right)\gamma +\left(1-\frac{no.DelayedPkts}{totalPketsRecv}\right)\delta$$

The first component (ω) of Equation (4) refers to the energy consumption on a particular path *P* and is calculated using Equation (5). *ω* is calculated from the energy consumed on a particular path until a critical point in the network, and afterwards, it calculates the value from the energy reserve along the path. The critical point in the network (Ω) is preset and is normally considered as 20% of the initial energy. If the minimum energy on the path *min.energy*(*P*) is less than Ω, the component ω is calculated from the power consumed along the path *P_w_*(*P*); otherwise, it is evaluated from the energy reserve in the path and is calculated using Equation (5).
(5)$$\omega =\{\begin{array}{c} min.energy\left(P\right),\;\;min.energy\left(P\right)<\Omega \;\\ \;P_{w}\left(P\right),\;\;\;\;min.energy\left(P\right)\geq \Omega \; \end{array}$$

The power consumption *P_w_*(*P*) is calculated from Equation (6); otherwise, it is equal to the minimum remaining energy calculated along path *P*.
(6)$$P_{w}\left(P\right)=\underset{d\in P}{\sum }E_{tx}\left(d\right)+\underset{s\in P}{\sum }E_{rx}\left(d\right)$$

$E_{tx}\left(d\right)$ and $E_{rx}\left(d\right)$ are the average values of power consumed during transmission and reception by the node along path P and calculated from Equations (1) and (2), respectively.

The second component (*min.buffer(p)*) of Equation (4) represents the minimum buffer of a node along the path *P*.

The third component, $\frac{1+maxHC-HC\left(P\right)}{maxHC}$ of Equation (4) estimates the number of hop counts. *maxHC* is the maximum hop count allowed in the network, whereas *HC*(*P*) is the hop count of the calculated path.

Finally,$\;1-\frac{no.DelayedPkts}{totalPketsRecv}$ is the fourth component that calculates the reliability of the path, that is, the ratio of packets delivered to the BS without delay to the total packets received.Step 7: Once the scores are calculated for the current set of solutions, the solution with the best score is saved and the same method is followed for each CH in the network.Step 8: Later, the optimization algorithm will process all the iterations and select a final route to transfer data from the CH to the BS.

## 4. Proposed Hybrid PSOGA

In this section, the routing algorithm for WSN that employs PSO and GA is discussed. This PSOGA method works in two steps. During the first phase, it obtains the population for a fixed number of epochs and holds the fittest M individuals while eliminating the rest of the individuals. During the second phase, various GA operators, namely selection, crossover, and mutation, generate the number of individuals equal to the ones omitted during the previous step. Further, these M individuals, in addition to the newly generated individuals, are used to generate the population of the next generation, thus integrating the advantages of PSO and GA. As a result, the proposed approach achieves a high convergence rate and global optima due to the PSO and GA, respectively. The proposed approach gradually increases the number of fittest individuals over each generation. Ahead of discussing the proposed hybrid approach, we discuss the PSO and GA approaches for further understanding.

### 4.1. PSO Algorithm-Based Routing 

PSO is a stochastic algorithm based on optimizing a candidate solution (or particle) [22,23,24,25,26]. It is a competent method used in numerous domains, such as architecture, research, education, and so forth. Owing to the proven effectiveness of PSO due to its efficiency, robustness, simplicity, and extreme ease of use, it is suited for various optimization problems. PSO particles travel in swarms inside a search space to find the optimum swarm solution by updating its location and speed, as shown in Figure 1.

In PSO, the swarm refers to the population, and the particle of the swarm corresponds to the solution. During the flight process, each particle moves in the problem space with a velocity that depends upon its previous position and the best position of the swarm. Now consider $x_{i}$ and $v_{i}$ are the position and velocity of the $i^{\mathrm{th}}$ particle, where the swarm has N particles. Here, particle i with a set of solutions is generally represented as $X_{1}=\left(X_{i1},\;X_{i2},X_{i3}\ldots ..X_{iN}\right)$. The position and velocity update of each particle during subsequent iterations is shown by the following equations [25]:(7)$$x_{i,m}\left(t\right)=x_{i,m}\left(t-1\right)+v_{i,m}\left(t-1\right)$$
(8)$$\;v_{i,m}\left(t\right)=w*v_{i,m}^{\left(t\right)}\left(t-1\right)+c_{1}*rand1\left(\right)*\left(pbest_{i,m}-x_{i,m}\left(t-1\right)\right)+c_{2}*rand2()*(gbest_{m}-x_{i,m}\left(t-1\right))$$

Here, m is the dimension of the solution space. *rand1* ( ) and *rand2* ( ) are the stochastic variables with uniform distribution and these variables are considered independent functions, that is, rand1 (),rand2 ()∈[0, 1]. $c_{1}$ and $c_{2}$ are the positive constants that refer to the cognitive and social constants, respectively. $pbest_{i,m\;}$is the best position that depends on the minimum path buffer, network lifetime, and hop count, with m solutions that are attained using the neighboring particles. Further, $gbest_{m}\;$indicates the global optimal.

PSO can also be employed for routing in a WSN by performing the following mathematical operations:

Particle Representation: A string of cluster heads is used to represent a particle that represents a feasible route in a swarm. Here, the CH makes a significant contribution to initializing the route during the swarm initialization. The BS generates a list $M_{i}$ with all single-hop cluster heads *j* for cluster head i. There must be a communication link from i to j.

Fitness Function: For each particle that represents a route, the fitness function is computed, indicating the maximum lifetime of the WSN. A WSN is alive until the energy of each CH along the route is exhausted. The fitness function L may be defined as [25] the following:(9)$$L=\frac{P\_initial}{P\_max}\;$$

Here, *L* indicates the network lifetime in terms of the number of rounds. P_initial and P_max refer to the initial energy and maximum energy, respectively, used during each round of communication.

Velocity: Velocity illustrates a binary operation that generates a new position for a cluster head. For instance, a velocity of (1,6) implies that the position of CH 1 will be updated to 6.

Position plus Velocity: if x and v indicate the position and velocity, respectively, the addition operation may be performed as follows [25]:x=(2, 7, 7, 4, 3, 5, 8)
v=(1, 6)(5, 5)

When performing x+v, for (1, 6), the new position of CH 1 will be 6, thus giving the results of (6, 7, 7, 4, 3, 5, 8). Further, after applying a velocity of (5, 5), we get (6, 7, 7, 4, 5, 5, 8).

Position minus Position: The minus operation for a position produces a velocity. Suppose route r1={2,4,5,6,8,8} and r2={2,6,5,7,8,8}. Then, the minus operation indicates the replaced cluster heads and is ((4,6),(6,7)).

Velocity plus Velocity: The addition of velocity $v_{1}$ and $v_{2}$ refers to the list of transpositions of CH in $v_{1}$, followed by those in $v_{2}$. With the addition of velocity, the identification of the CH never has a copy in the resultant velocity.

Coefficient times Velocity (Multiplication): Consider that m and v represent the coefficient and velocity, respectively. Here, velocity v=(i,j)|i∈{1,2…N}. Now, *m* times *v* yields another velocity, $v^{\prime }=\left(i,j\right)|\;i\in \left\{1,2\ldots N\right\}$ and $j\;\in \;N_{i}$ (neighbor of CH i).

### 4.2. Genetic Algorithm-Based Routing 

GA is a search-based optimization approach that employs the principle of genetics and natural selection [27,28,29]. It is often used to provide optimal or suboptimal solutions to challenging real-life problems that might take a longer time to resolve. The steps performed for applying a GA to solve an energy optimization problem in a WSN are given as follows:Step 1: Initially, the chromosome is encoded in an efficient manner.Step 2: In order to maximize the network lifetime, the individual with the highest fitness function value is taken in the upcoming generation.Step 3: The mating pool of good individuals is created through selection.Step 4: Among the pool of good individuals, two parents are selected to exchange their genes to create new offspring. The newly created offspring are expected to have better fitness values than the parents.Step 5: Mutation is also employed to achieve diversity in the offspring generated during the crossover.

### 4.3. PSOGA-Based Routing Algorithm 

The genetic algorithm uses its individuals to find the local solution that feeds the PSO. The PSO is only able to find the global solution since the process of finding the local solution, that is, the local best solution, is accomplished by the GA. The main aim is to reduce the premature convergence by the PSO that falls in the local optima. Hence, to obtain a developed solution, we mainly considered the formation of the local solution by the genetic individuals or chromosomes. Here, each individual represents a potential solution. Furthermore, in the GA, each particle is regarded as an individual chromosome, and the swarm refers to the entire population.

The PSOGA starts with the process of generating random individuals while considering the total iterations as a parameter for the algorithm. The population aims to provide solutions to the route planning, and the solution is considered in a distributed manner over the whole IoT network. The initial population is allowed to pass through the GA in its initial iterations. This helps in reducing the route selection score, as the GA performance entirely depends on encoding the solutions in particles and chromosomes. In addition, it considers the measurement of the fitness function, population size, and the total number of iterations. Such parameters are adjusted after the evaluation of the GA on the initial trails. The PSO starts its operation after obtaining the local solutions by the GA during the initial iterations. The PSO uses particles to find the global solutions that represent the overall solutions in finding the optimal routes. The steps are as follows:Step 1: Initialize a swarm of particles with a random position and zero velocity.Step 2: Evaluate the objective values of the particles.Step 3: Evaluate the fitness function value for each particle.Step 4: Determine the local and global best solution.Step 5: Until the termination condition is reached, update the velocity and position of each particle, and update the local and global best solution.Step 6: Arrange individuals in decreasing order of fitness function value and find the *M* best particles.Step 7: GA Evolution—reproduce *Pop_size-M* (performing subtraction) and GA individuals, and implement crossover and mutation operators to create *Pop_size-M* particles.Step 8: Combine and form *Pop_size* individuals.Step 9: End.

The application of PSOGA is followed by updating the V* (see Table 1). Further, equation (4) is used to evaluate the route score, followed by a zigzag scan that starts in the upper-left corner of the grid. It sequentially scans the diagonals of the grid to determine the route score. Once the evaluation has been done, the CH communicates the same to the neighboring grid. The same pattern is followed subsequently for the remaining grids. The present work was focused on static WSN nodes. In case a CH fails, there is a packet drop in the network; all the parameters will be reset and the re-election of the CH will begin.

### 4.4. Trust-Based Cluster Head Selection 

Over the past few decades, researchers have employed a trust parameter in WSNs [30,31,32,33,34]. For instance, the authors of [35] proposed a framework that forms clusters in a manner so as to prevent the likelihood of compromised or malicious nodes being elected as CHs. For the same purpose, the current model uses statistical methods to evaluate trust without considering the trust among the sensor nodes. In [35], the authors suggested a model that evaluates trust using a classical probability model. Furthermore, the suggested model employs simple statistical methods for the same purpose, barring the trust recommendation among sensor nodes that prevents nodes from reflecting trust in an accurate manner.

The authors of [36] also proposed a trust-based LEACH (T-LEACH) protocol that utilizes CH-assisted monitoring to reduce energy consumption. In the proposed model, the trust module consists of two different modules, namely, a monitoring module and a trust evaluation module. Here, the monitoring module observes the network and reports to the trust evaluation module if any misbehavior is noticed. Further, a node also maintains a neighbor situational trust table (NSTT) loaded with a trust value for each pair of node IDs and situation operations, such as data sensing, localization, and routing. As a result, T-LEACH loses less data than LEACH, as half of all the information sent by cluster participants is processed by the gateway. However, it is not possible to stop the constant loss of data in T-LEACH because of the lack of monitoring on the cluster head. 

Similar work is also carried out in [37] by the proposed node behavioral techniques trust analysis algorithm banding belief theory (NBBTE), which combines the method of the node behavioral strategy and Dempster–Shafer (DS) evidence theory. Here, a variety of network application-related trust factors, such as the packet size, the forwarding rate of the packet, data reliability, and security grade and coefficients are considered to determine the confidence values by measuring the weighted average of the trust factors. The simulation tests demonstrated that the proposed scheme is capable of effectively evaluating the trust nodes. However, the method of trust evaluation can entail excess energy, time, and costs due to the cooperation and interaction with neighbors and other numerous parameters.

In [19], a cluster head selection method was proposed, which is based on a trust factor that ensures all nodes are trustworthy and authentic during communication. To achieve this, direct trust is calculated using parameters such as the residual energy and the distance between the nodes, along with the use of the k-means clustering algorithm. The selection factor, *F*(*f*), for recommended nodes is evaluated using the following weight function:(10)$$F\left(f\right)=w_{1}*\left(\frac{e}{d}\right)+w_{2}*trust\_value\;\;$$
(11)$$w_{1}+w_{2}=1\;\;$$
where $w_{1}$ and $w_{2}$ are the weight values, which can vary between 0 and 1. The sum of $w_{1}$ and $w_{2}\;$should be equal to 1. e is the energy and d is the distance. 

After measuring the fitness function, the node with the highest fitness value is selected as the cluster head.

One sensor node in each cluster must act as the CH. The CH depletes its energy faster compared to the other nodes. Hence, the role of the CH must be periodically rotated among different sensor nodes to achieve a longer lifetime of the WSN. The CH node aggregates the results from nodes in the cluster to recognize events in the region of interest. Finally, the CH reports such events to the sink node. Further, the sink node determines the average node energy for all the nodes in a network. Now, the nodes with less energy than a threshold will not be selected as the CH during subsequent rounds. This will enhance the network lifetime by distributing the CH role uniformly among all the networks. See the flowchart in Figure 2 for the cluster head selection process.

### 4.5. Proposed Hybrid-Based PSOGA Routing 

The proposed algorithm works with two important criteria—the selection of cluster heads using the trust mechanism and finding the routing path using the PSOGA. Details about the selection of the cluster head and cell headers using a trust-based mechanism are given in Section 4.4, and the set of routing paths using the PSOGA is given in Section 4.3. The parameters used in the algorithm description are given in Table 1.

The algorithm for the proposed model is given in Algorithm 1 as follows:
**Algorithm 1** Proposed Trust-Based PSOGA Algorithm1.*For i = 1 to number of grids* *While Fch ≠ S*  *Generate V* using zigzag method*2.*BestScore = 0;*3.*For it = 1 to Max_Iterations*  *For i = 1 to rows (V*)* *Temp = V*(i, Na)*  *For j = 1 to length (Temp)*  *For k = Temp(j)*  *extract score factors from Ir(Ni)**End for* *End for*  *R_score(i)_= Score_(i)_;//from equation 4 calculated*  *End for*  *G__BestScore, index_ = max (R_score_);*  *if L__BestScore_ > G__BestScore_*  *G__BestScore_ = L__bestScore_;*  *SelR = V*(index,:);*  *End if*  *Update V* using PSOGA Algorithm**End for*  *End while*  *End for*4.*Perform Communication between CH to sink using selected final route*5.*Evaluate Network performance*6.*Stop Algorithm*

In the proposed algorithm, Step 1 generates the random routes for each grid in the network. The process of determining the random route continues while the farthest CH is not equal to the sink node. Step 2 initializes the best score to 0. Step 3 updates the path V* determined in Step 1. For this, it uses the concept of the local score and global score. Step 4 initiates communication from the CH to the sink node through the path obtained in Step 3. The performance of the network is evaluated as per Step 5 to obtain the performance metrics, such as the longevity of the nodes, the network lifetime, the packet delivery ratio and throughput, and so forth. The most prevalent genetic algorithm (GA) and swarm intelligence technique (PSO) benefits are taken into account for getting a better convergence and path to improve the network lifetime. 

## 5. Network Setup

The simulation was performed under the university’s central server; the authors used multi-hop networks with various numbers of homogenous sensor nodes. The region under consideration was rectangular and divided into a number of grids. This is in line with the majority of the published literature wherein the deployed network is considered to be rectangular. The BS was located at the center or at one of the edges. A rectangular grid of sensor nodes was used for relaying packets and the model for transmission was considered to be fixed power transmission. In this scenario, the authors simulated different routing approaches, namely PSOGA (proposed), LEACH, and PSO. The simulation of different approaches enabled the authors to establish the efficacy of the proposed approach vis-a-vis state-of-the-art methods. As per the proposed method, the sensor nodes were assumed to have an initial fixed energy and the BS had unlimited energy. The BS also had knowledge of the deployed nodes. The basic structure of the network is shown in Figure 3, in which the number circled in each grid represents the CH, and the green cross mark represents the BS.

This grid-based approach is popularly known for its features, namely uniformity in energy consumption, scalability, and simplicity [20,33,38]. However, there are some major challenges associated with a grid-based approach, such as the determination of the optimal number of grids for a particular scenario, a non-uniform grid size, and the improper selection of the CH [39]. These challenges further worsen owing to the dynamic behavior of WSNs. The authors of this paper primarily focused on developing a robust model that can handle various issues such as heterogeneity in node density, deployment area, and grid size [38,40,41].

The applicability of our strategy is immense, for example, vast horticultural fields where environmental parameters, such as atmospheric temperature and humidity, soil moisture, soil temperature, and soil pH, are to be measured [42]. In addition, it can be applied to any event detection application, such as wildfire detection [43] and seismic monitoring [44]. Apart from these, it can also be used in structural health monitoring where parameters such as humidity, temperature, stress, and strain are required to be measured [45]. In several of these above scenarios, it is cost effective to use non-rechargeable nodes. 

In real-life deployments, the geographical area may not be rectangular. Our thinking is that a rectangle can be fitted to any irregular geographical area by using its extents, much like how irregular features in images are processed in a bounded rectangle. This is likely to have the effect of having some grids on the periphery of the geographic region where there may be very few nodes, or even none. An empty grid will not require a cluster head and will, thus, not participate in the path selection process.

It is assumed that a large number of radio channels are available for transmission, which will mitigate interference issues [46].

In Section 6, the evaluation of the PSOGA is presented with various performance metrics, including the total residual energy, the network throughput, the total number of alive nodes at the end of each iteration, and the packet delivery ratio (PDR). The PSOGA was compared with the conventional LEACH protocol, with the parametric setting given in Table 2.

## 6. Results and Discussion

The proposed strategy was further tested in five different scenarios with varying network area sizes, number of grids/clusters, and total number of nodes, as shown in Table 3, in which the number of grids varies with different cases and ranges between 8 and 100, with the node population ranging from 100 to 1250 based on the size of the network area. The details of these scenarios are given in Table 3.

The entire setup was in an area with the base station located at the center, as shown in Figure 3.

The proposed PSOGA method was compared with the conventional LEACH protocol. The parametric settings are given in Table 2 and the grid formation is shown in Figure 3. The performance evaluation was done based on the total number of packets reaching the BS, the total number of alive nodes, the residual energy of the network, the packet delivery ratio (PDR), and the throughput of the network. The packet delivery ratio (PDR) indicates the ratio between the number of packets that the sink or destination receives and the entire number of packets sent by the source. Section 6.1, Section 6.2, Section 6.3, Section 6.4, Section 6.5, Section 6.6 and Section 6.7 demonstrate the performance evaluation based on different grid configurations and network settings. For each scenario, the simulation was performed ten times and the mean was plotted for each metric.

### 6.1. Case Study 1 

Figure 4 shows the results of Case Study 1: 2 × 4 grids, 100 nodes. The network structure (Figure 4a) shows eight grids with the selected cluster heads marked in numbers in triangular shapes and connected by black lines. The comparative results of the performance metrics are shown in Figure 4b–f. For each metric, the network was generated and the simulation was performed 10 times. The average of the 10 simulations, each over 2500 rounds, were plotted, along with the standard deviation. 

The number of packets reaching the BS is shown in Figure 4b. After 2500 rounds, the packets sent to the BS with the PSOGA improved by 278.94% and 5.55% as compared to LEACH and PSO, respectively. Figure 4c shows the number of nodes alive at each round. The points at which all the nodes were dead for LEACH, PSO, and the proposed PSOGA were after 700, 1900, and 2000 rounds, respectively, which shows significant improvements of 185.7% and 5.26% over LEACH and PSO, respectively. The residual energy also demonstrated promising improvement, as shown in Figure 4d. The energy was depleted after 500, 1700, and 1800 rounds for LEACH, PSO, and the proposed PSOGA, respectively. Thus, the network energy in the proposed PSOGA achieved an enhancement of 259.06% compared to LEACH and 9.97% compared to PSO. Figure 4e demonstrates improvements in the PDR by 298.75% and 67.5% compared to LEACH and PSO, respectively. Furthermore, the throughput with the PSOGA was increased by 7.13% and 49.5% as compared to PSO and LEACH, respectively, as shown in Figure 4f.

Figure 4g is a representative graph showing the nodes alive for each of the 10 simulations for LEACH, PSO, and PSOGA. The average of the 10 simulations is shown in Figure 4c. As mentioned earlier, 10 simulations were similarly performed for each of the other metrics. The bar graph in Figure 4h shows the simulation times for LEACH, PSO, and PSOGA. Figure 4i shows the difference between PSO and PSOGA, from 1350 to 1650 rounds.

### 6.2. Case Study 2 

Figure 5 shows the results of Case Study 2: 4 × 4 grids, 100 nodes. The network structure (Figure 5a) shows 16 grids, with the selected cluster heads marked in numbers and connected by a black line. The results of the performance metrics show that with 100 nodes, the packets sent to the BS (Figure 5b) improved by 268.42% and 9.36% when compared to LEACH and PSO, respectively. Further, the PSOGA demonstrated an increase in the number of alive nodes by 155.43% and 10% as compared to the PSO and LEACH, respectively. The results in Figure 5d illustrate that the number of alive nodes became zero after 500, 1850, and 1994 rounds for LEACH, PSO, and the proposed PSOGA algorithm, respectively, leading to significant gains of 274.64% and 10.4% vis-à-vis LEACH and PSO, respectively. Similarly, the PDR was improved by 314.25% and 72.48% in comparison to LEACH and PSO, respectively (Figure 5e). Finally, the proposed PSOGA demonstrated throughput enhancement of 37.5% and 4.84% compared to LEACH and PSO, respectively, as shown in Figure 5f.

As compared to Case study 1: 2 × 4 grids, 100 nodes, it is evident that although the number of nodes in both the case studies was the same, the difference lies in the grid structure. In the case of the 4 × 4 grid, the network performed better in terms of the evaluated parameters. For instance, the number of packets sent to the BS in Case Study 2 was 3.5 × 10^6^, almost 3 × 10^5^ higher than in Case Study 1, which had 3.2 × 10^6^ packets. Similarly, the network structure demonstrated a slight improvement in the throughput, with a value of almost 17,500 as compared to 16,000 in the 8-grid structure.

### 6.3. Case Study 3 

Figure 6 shows the results of Case Study 3: 4 × 4 grids, 200 nodes. The network structure (Figure 6a) shows 16 grids with the selected cluster heads marked in numbers and connected by a black line. The results of the performance metrics show that with 200 nodes, the packets sent to the BS (Figure 6b) with the PSOGA improved by 288.7% and 9.36% compared to LEACH and PSO, respectively. Furthermore, the number of alive nodes improved by 189.44% and 11.65% as compared to LEACH and PSO, respectively (Figure 6c). Figure 6d also demonstrates a similar trend with respect to residual energy, that is, the energy of the network was reduced to 0 after approximately 500 rounds, 1850 rounds, and 1994 rounds for LEACH, PSO, and the proposed PSOGA, respectively, contributing to improvement percentages of 279.2% and 12.5%, respectively. Similarly, the PDR was also enhanced by 304.9% and 67.3% compared to LEACH and PSO, respectively (Figure 6e). Figure 6f further demonstrates the efficacy of the PSOGA in terms of throughput improvement by 44.53% and 5.96% compared to LEACH and PSO, respectively.

We compared the performance metrics of Case Study 3 to Case Study 2, in which the number of grids was the same, that is, 16, but the number of nodes was higher, that is, 200 nodes. The number of packets reaching the BS in Case Study 3 was higher, with 3.6 × 10^6^ vs. 3.5 × 10^6^ packets, demonstrating a slight improvement of around 1 × 10^5^ packets. Moreover, in Figure 5f and Figure 6f, minimal improvement in the throughput can be seen.

### 6.4. Case Study 4

Figure 7 shows the results of Case Study 4 with 10 × 10 grids, 625 nodes. The network structure (Figure 7a) shows 100 grids with the selected cluster heads marked in numbers and connected by a black line. The results of the performance metrics show that with 625 nodes, the packets sent to the BS (Figure 7b) saw improvements of 325% and 30.76% as compared to LEACH and PSO, respectively. The number of alive nodes was enhanced by 232.22% and 6.26% as compared to LEACH and PSO, respectively. In Figure 7c, the residual energy improved by 298.4% and 7.67% with respect to LEACH and PSO, respectively, as shown in Figure 7d. Furthermore, in Figure 7e, the PSOGA demonstrated increases of 266.36% and 46.47% in the PDR as compared to LEACH and PSO, respectively. Similarly, Figure 7f shows increases in the throughput of the network by 46.3% and 6.87% when compared to LEACH and PSO, respectively.

In Case Study 4, with 625 nodes and 100 grids, the performance of the network deteriorated as compared to the previous case studies. The number of packets sent to the base station was 3.1 × 10^6^, which is substantially lower than in the previous case studies. Similarly, the PDR was also reduced to 60% from the 70% achieved in Case Study 3. The throughput achieved in Case Study 4 was 15,500, much less than the 18,000 achieved in Case Study 3. The drastic reduction in the performance metrics can be attributed to the structure, where, although the number of nodes was higher, the grid structure of 10 × 10 made the process of path discovery cumbersome and, hence, reduced the throughput of the network.

### 6.5. Case Study 5

Figure 8 shows the results of Case Study 5 with 10 × 10 grids, 1250 nodes. The network structure (Figure 8a) shows 100 grids, with the selected cluster heads marked in numbers and connected by a black line. The results of the performance metrics show that with 1250 nodes, the packets sent to the BS (Figure 8b) in the PSOGA achieved improvements of 333% and 45.83%, respectively, as compared to LEACH and PSO. Furthermore, Figure 8c illustrates increases in the alive nodes by 236.7% and 5.89% when compared to LEACH and PSO, respectively. The PSOGA showed significant improvements in the residual energy by 301% and 7.9% compared to LEACH and PSO, respectively (Figure 8d). PSOGA also showed substantial improvements in the PDR by 274.7% compared to LEACH and 46.92% compared to PSO (Figure 8e), and the throughput of the network with the PSOGA increased by 54.82% and 6.98% as compared to LEACH and PSO, respectively (Figure 8f).

In Case Study 5, with 1250 nodes and 100 grids, the performance of the network improved as compared to Case Study 4. The number of packets sent to the base station was 3.2 × 10^6^, which is higher than in the previous case study. Similarly, the PDR was also enhanced to 65% from the 60% achieved in Case Study 4. The throughput achieved in Case Study 5 was 17,000, higher than the 15,500 achieved in Case Study 4. The enhancement in the performance metrics can be attributed to the structure, where, although the number of grids is higher, the higher number of nodes—1250—sustained the network for a longer time and, thus, enhanced the performance metrics of the network.

### 6.6. Comparative Analysis

The results over various node densities and varying network grids were analyzed and the comparative data of the network lifetime are presented in Figure 9. The first node dead (FND), half node dead (HND), and last node dead (LND) were the metrics of comparison between LEACH, PSO, and the PSOGA. In all cases, the PSOGA showed significant improvement over LEACH for all the metrics. The PSOGA also showed improvements over PSO, albeit to a much lesser extent than the improvements compared to LEACH.

### 6.7. BS at the Edge

To complete the study and evaluate the efficacy of the proposed algorithm, the BS was placed at the edge of the network, as shown in Figure 10a. This case study enabled the evaluation of the performance of the PSOGA when the location of the base station is at the edge of a 2 × 4, 100 nodes configuration. In this case study, the PSOGA also showed notable improvements in various performance metrics. Figure 10b depicts the significant improvement in the number of packets sent to the BS, with the PSOGA exhibiting 278.94% and 9.0% improvements compared to LEACH and PSO, respectively. As shown in Figure 10c, PSOGA achieved improvements in the number of alive nodes by 192.93% and 12.47% as compared to LEACH and PSO, respectively. Furthermore, Figure 10d demonstrates the enhancements of 281.4% and 611.85% in the residual energy for PSOGA compared to LEACH and PSO, respectively. Similarly, substantial enhancements in the PDR by 302.96% and 66.28% for the PSOGA compared to LEACH and PSO, respectively, are visualized in Figure 10e. Additionally, Figure 10f depicts the improvements of 45.87% and 6.54% in the throughput of the network with the PSOGA compared to LEACH and PSO, respectively. The plotted curves are the averages of the data from 10 simulations.

## 7. Conclusions and Future Scope

In this study, a trust-based PSOGA model was developed to increase the network lifetime in an IoT-based WSN environment. In this model, after the formation of grids, the optimal selection of the CH using a trust model among the stationary nodes enables reliable data transmission from the sensor nodes to the cluster head. After forming the initial random paths using a zigzag strategy, the optimization algorithm helps in finding the best routes that offer the quickest delivery of data packets to the sink node. The consideration of metrics, such as route score, number of received packets, number of delayed packets, residual energy, power consumed, and total hop counts, enables the optimal selection of routes among the random paths. Such routes are reliable enough to carry out the packet transmission, allowing for higher residual energy. The use of the PSOGA ensures the faster selection of routes than the existing LEACH mechanism; the proposed PSOGA obtained increased network lifetimes in all five different case studies. The percentages of dead nodes in the network after several iterations were significantly lower than those of the LEACH protocol, which shows the efficacy of the present system. 

Currently, the authors are engaged in extending the presented work in the direction of data reduction, coupled with security, to further enhance the network efficiency and privacy. In the future, deep learning algorithms can be deployed for bufferless systems that match the speed of input data acquisition from several IoT devices. In addition to this, further research will consider a cross-layer approach to increase the network efficiency and lifetime when faced with node failures.

## Figures and Tables

**Figure 1 sensors-21-07439-f001:**
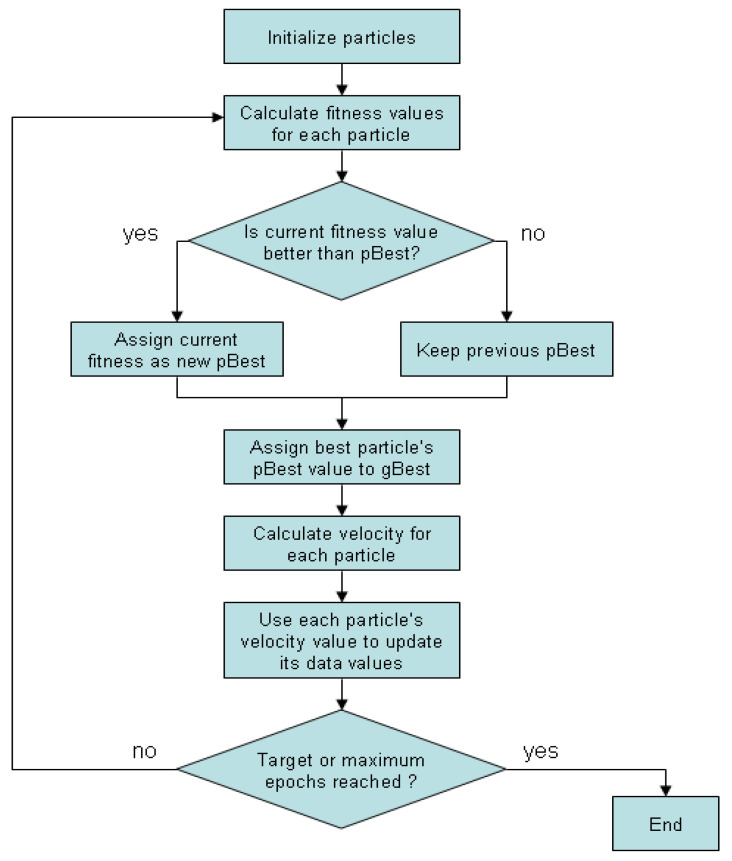
Flowchart of the PSO algorithm.

**Figure 2 sensors-21-07439-f002:**
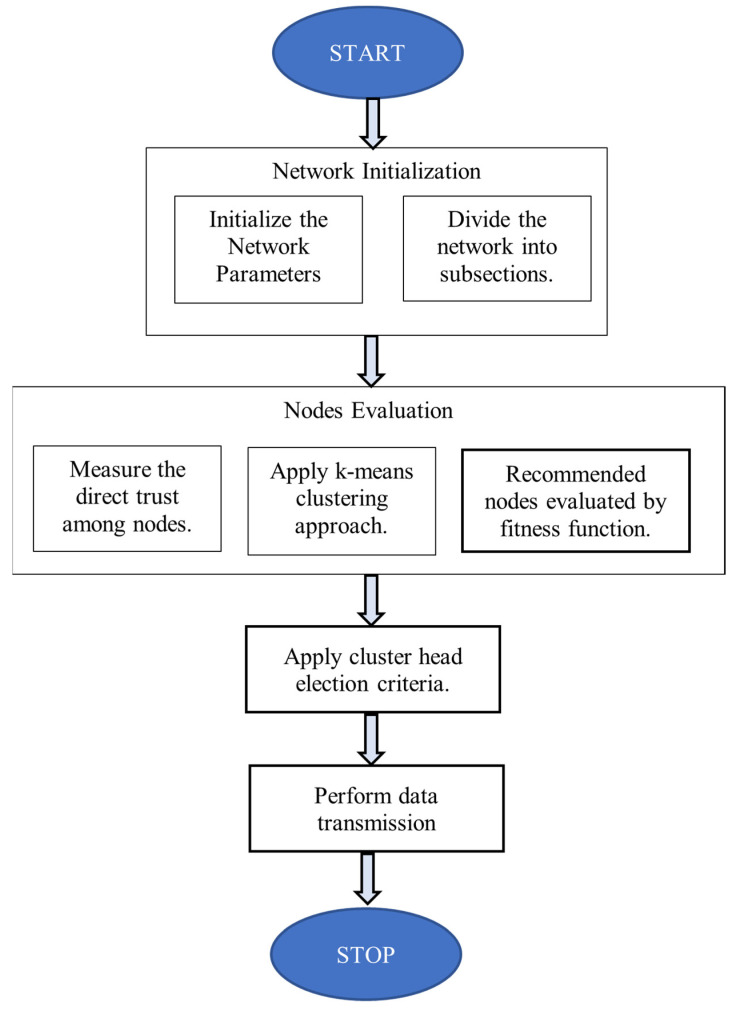
Flowchart for cluster head selection.

**Figure 3 sensors-21-07439-f003:**
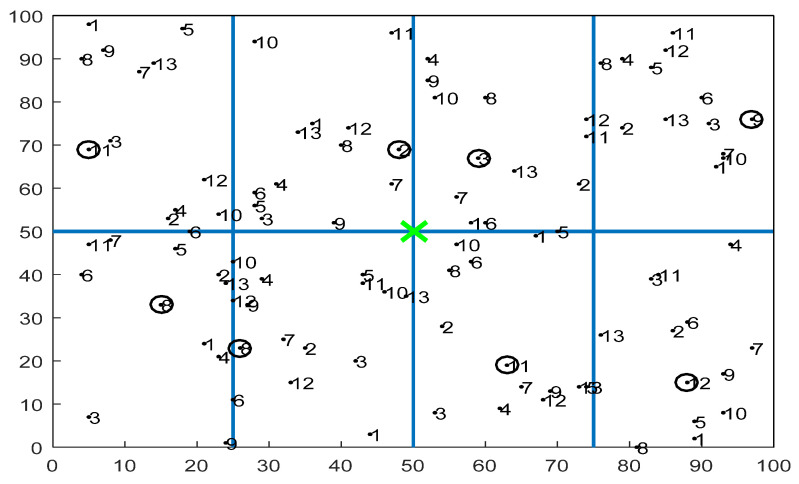
Grid formation in a WSN.

**Figure 4 sensors-21-07439-f004:**
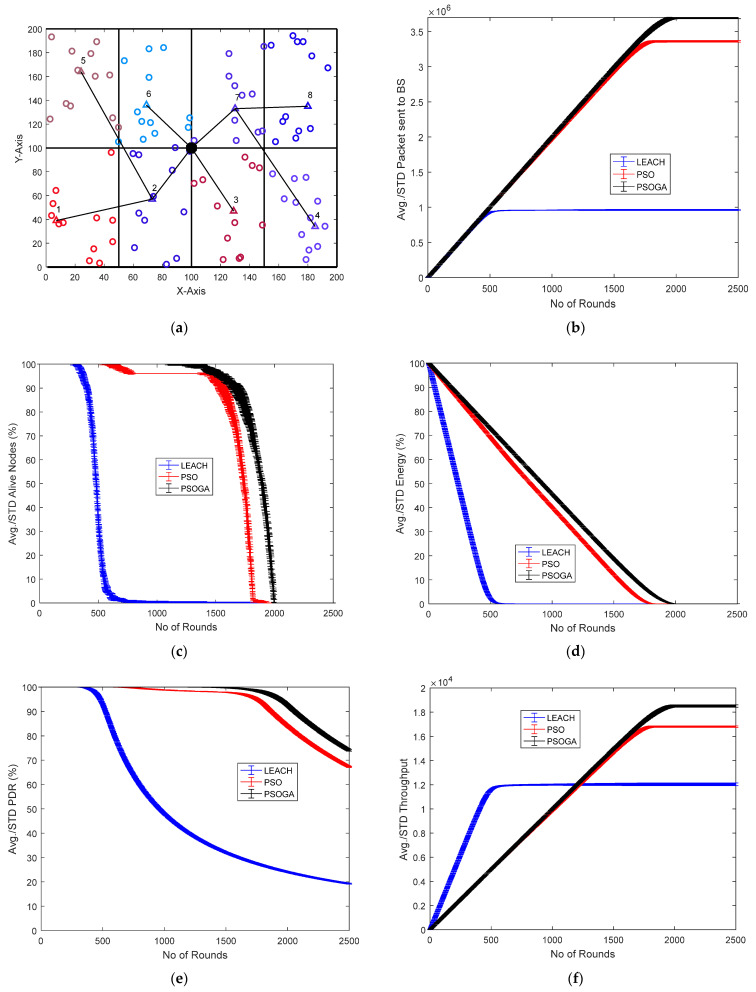
Results of Case Study 1: 2 × 4, 100 nodes. (**a**) Network structure for Case Study 1. (**b**) Packets reaching the BS. (**c**) Total alive nodes. (**d**) Total residual energy. (**e**) PDR. (**f**) Throughput. (**g**) Graph showing data plots of 10 simulations. (**h**) Simulation time for 10 simulations. (**i**) Error bound for alive nodes.

**Figure 5 sensors-21-07439-f005:**
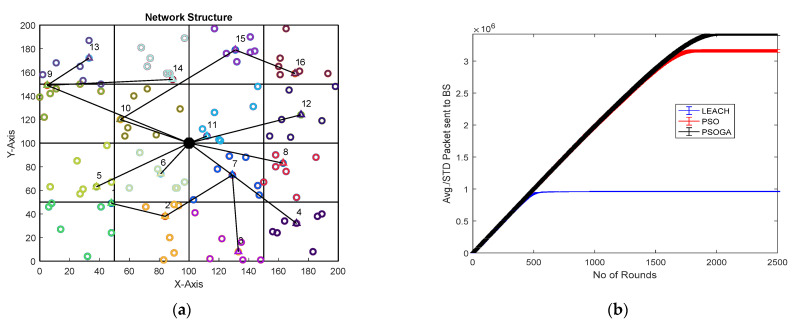
Results of Case Study 2: 4 × 4, 100 nodes. (**a**) Network structure for Case Study 2. (**b**) Packets reaching the BS. (**c**) Total alive nodes. (**d**) Total residual energy. (**e**) PDR. (**f**) Throughput.

**Figure 6 sensors-21-07439-f006:**
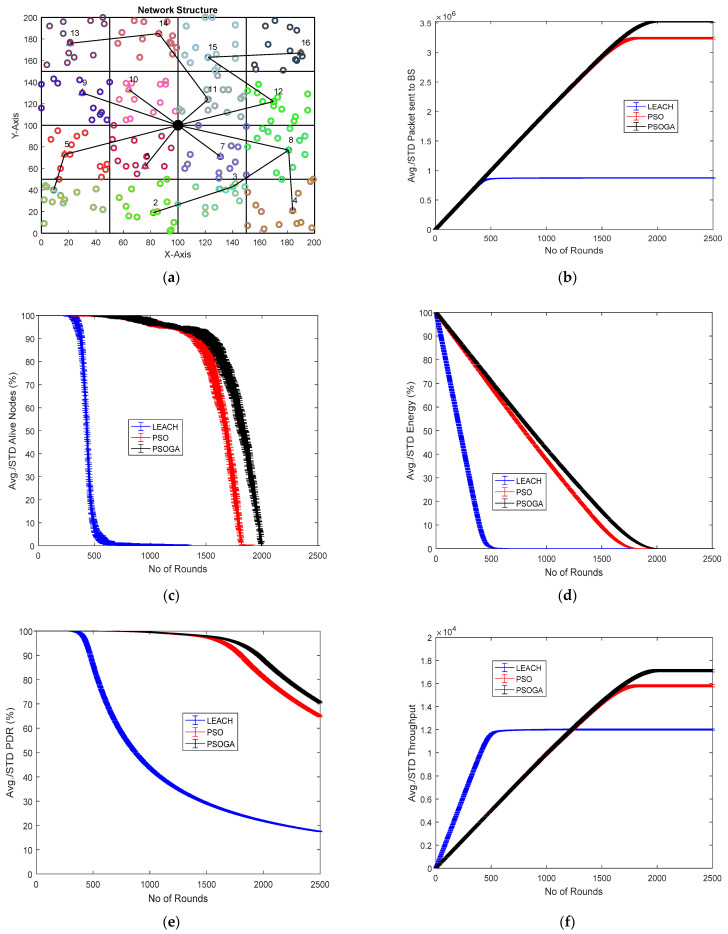
Results of Case study 3: 4 × 4, 200 nodes. (**a**) Network structure for Case Study 3. (**b**) Packets reaching the BS. (**c**) Total alive nodes. (**d**) Total residual energy. (**e**) PDR. (**f**) Throughput.

**Figure 7 sensors-21-07439-f007:**
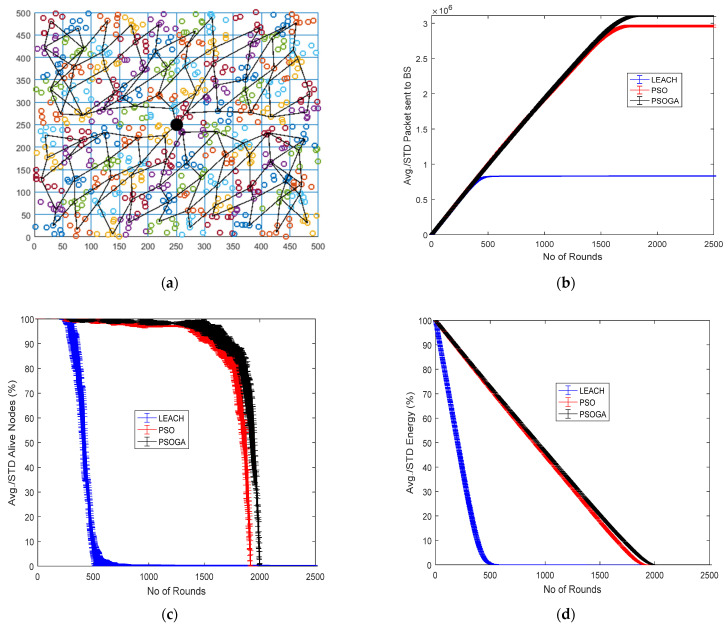
Results of Case Study 4: 10 × 10, 625 nodes. (**a**) Network structure for Case Study 4. (**b**) Packets reaching the BS. (**c**) Total alive nodes. (**d**) Total residual energy. (**e**) PDR. (**f**) Throughput.

**Figure 8 sensors-21-07439-f008:**
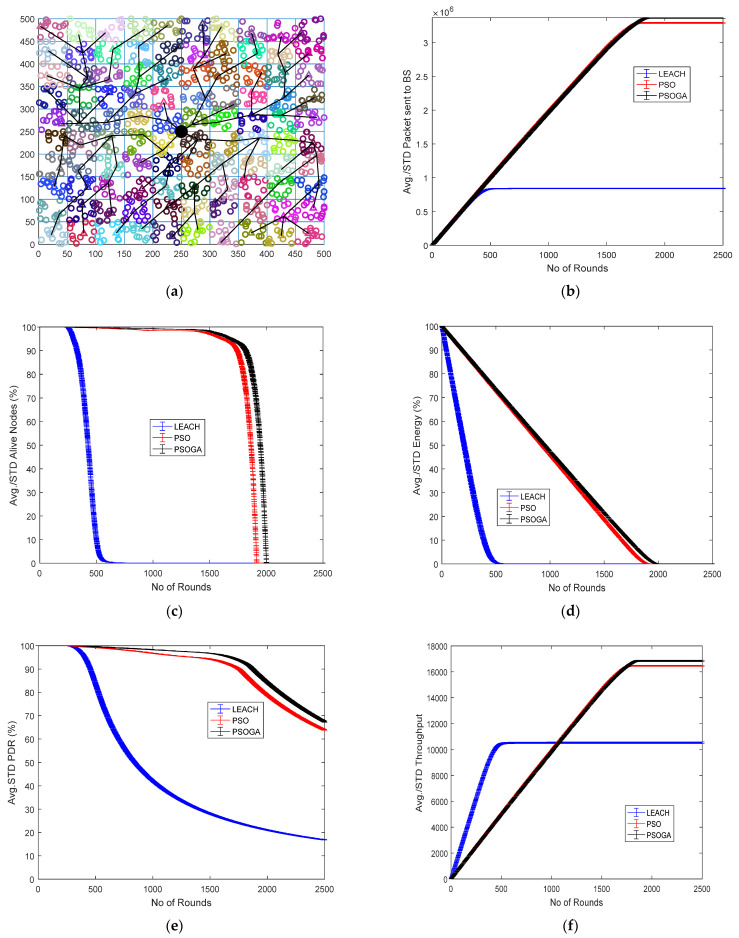
Results of Case Study 5: 10 × 10, 1250 nodes. (**a**) Network structure for Case Study 4. (**b**) Packets reaching the BS. (**c**) Total alive nodes. (**d**) Total residual energy. (**e**) PDR. (**f**) Throughput.

**Figure 9 sensors-21-07439-f009:**
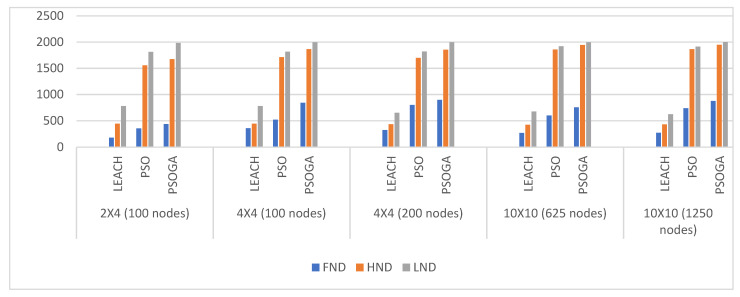
Overall lifetimes of sensor nodes, comparing the first node dead (FND), half node dead (HND), and last node dead (LND) between the proposed and existing models.

**Figure 10 sensors-21-07439-f010:**
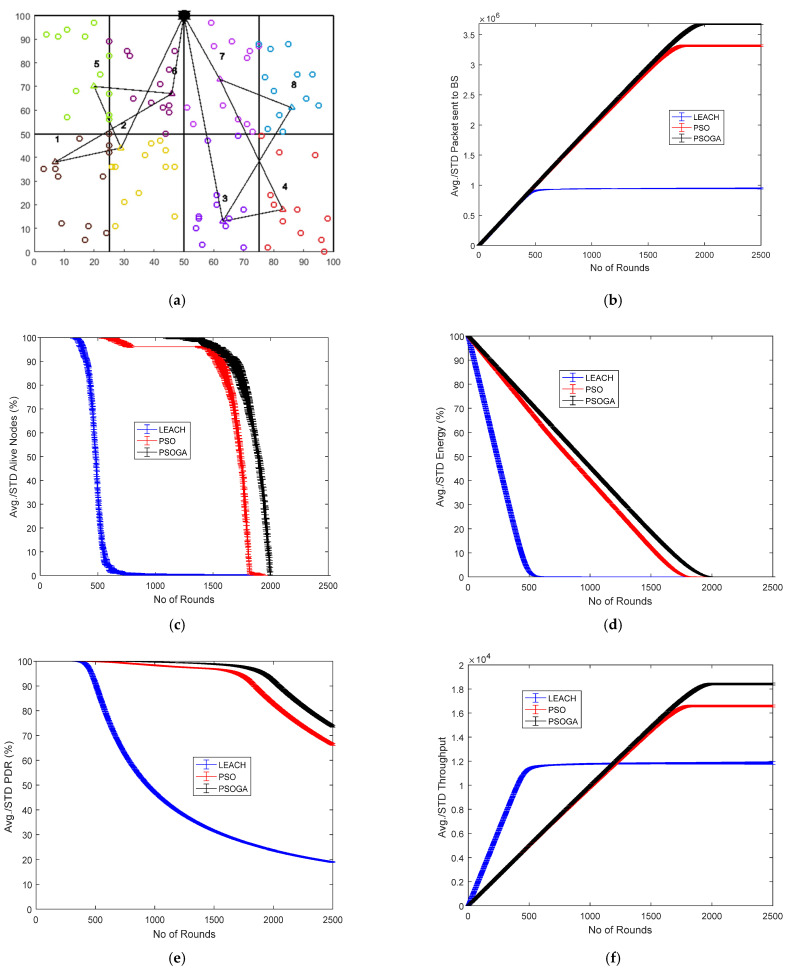
Results of BS at edge, 2 × 4 100 nodes. (**a**) Network structure BS at edge. (**b**) Packets reaching the BS. (**c**) Total alive nodes. (**d**) Total residual energy. (**e**) PDR. (**f**) Throughput.

**Table 1 sensors-21-07439-t001:** Parameters used in the proposed PSOGA algorithm.

Parameter	Description
Ir(N_i_)	Information register with information of individual node Ni in the network
A	Area
N_i_	Node i
L_s_	Sink location
N_s_	Number of grids in a network
F_ch_	Farthest CH
S	Network sink
V*	Set of random routes of a grid to adjacent grids in the network. For example, from the farthest CH to the sink from (F_ch_ to S)
BestScore	Best route score
N_a_	All nodes in route
R_score (i)_	Global route score
Score _(i)_	Individual route score
G_BestScore_	Global best score
L_bestScore_	Local best score

**Table 2 sensors-21-07439-t002:** Network parameter setup.

Parameters	Values
Network area	100 m × 100 m, 200 m × 200 m and 500 m × 500 m
Total number of nodes	100, 200, 625 and 1250
Initial energy	0.2 J
Power amplification ($\varepsilon_{fs}$)	10 pj/bit/m2
Power amplification ($\varepsilon_{mp}$)	0.0013 pj/bit/m4
Transmitter/receiver energy (*E_elec_*)	50 nJ/bit
Base station location	(50, 50), (125, 125) and (250, 250)
Number of rounds	2500

**Table 3 sensors-21-07439-t003:** Different network setups for simulation.

Case Study	Network Area	Number of Grids	Total Number of Nodes
1	100 m × 100 m	2 × 4	100
2	100 m × 100 m	4 × 4	100
3	200 m × 200 m	4 × 4	200
4	500 m × 500 m	10 × 10	625
5	500 m × 500 m	10 × 10	1250

## Data Availability

No new data were created or analyzed in this study. Data sharing is not applicable in this article.

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
