# Peer review of "Network Lifetime Improvement through Energy-Efficient Hybrid Routing Protocol for IoT Applications"

_sensors, 2021, doi:10.3390/s21227439_

Round 1

Reviewer 1 Report

The authors made a significant improvement to the manuscript. They addressed all the issues from my previous reviews, apart from one. Figures in Figure 4 show average as one line and a standard deviation as another line. This is not the correct way how to display standard deviation. An example of a correct way can be seen here:

or here

https://www.researchgate.net/profile/Shuxiang-Fan-2/publication/321432382/figure/fig2/AS:715140450250759@1547514178601/Mean-reflectance-spectra-solid-line-and-standard-deviation-error-bar-before.png

Additionally, figures 5, 6, 7, 8, and 10 should similarly depict average with standard deviation.

Author Response

Reviewer Comments

  • The authors made a significant improvement to the manuscript. They addressed all the issues from my previous reviews, apart from one. Figures in Figure 4 show average as one line and a standard deviation as another line. This is not the correct way how to display standard deviation. An example of a correct way can be seen here:

or here :https://www.researchgate.net/profile/Shuxiang-Fan-2/publication/321432382/figure/fig2/AS:715140450250759@1547514178601/Mean-reflectance-spectra-solid-line-and-standard-deviation-error-bar-before.png

Response 1: We are thankful to the reviewer for the comments and helping us by providing an example of the figure. Based on the provided example, mean and standard deviation are plotted for all the scenarios.

  • Additionally, figures 5, 6, 7, 8, and 10 should similarly depict average with standard deviation.

Response 2: Now all the figures in the present manuscript depict the same.

Reviewer 2 Report

In this new version, the authors have addressed my previous concerns. Therefore, I consider the paper suitable for publication.

Author Response

Reviewer 2 Comments

  • In this new version, the authors have addressed my previous concerns. Therefore, I consider the paper suitable for publication.

Response 1: We are thankful to the reviewer for giving us valuable comments.

This manuscript is a resubmission of an earlier submission. The following is a list of the peer review reports and author responses from that submission.

Round 1

Reviewer 1 Report

This paper proposes a new algorithm for generating paths for delivering data from source nodes via cluster heads to the base-station. Cluster heads are generated using a trust model and then the proposed algorithm, combinig Particle Swarm Organisation (PSO) and Genetic Algorithms (GA) is used to generate paths.

The paper is generally well structured and fairly easy to read. The main drawback of the paper is its limited contribution and missing some information. The authors used a generally known PSO algorithm to generate paths and just applied Genetic Algorithm to slightly improve the paths generated by PSO. The authors also make an unsubstituted claim that the paths generated by their PSOGA algorithm is "optimal" or that the algorithm finds the "best routes". There is no proof of that. The paper just shows that the paths are better than the paths generated by PSO.

The energy model of their algorithm seems to assume a possibility of continuous change of transmission power. However, this is not the case in real life, where sensor nodes have discrete settings of transmission power. The model does not seem to take into account radio interference.

It is also not clear, where the algorithm is executed. Is it at the base-station? Does the base-station need to have full knowledge of the deployed WSN?

The description of the algorithm on page 11 is very hard to follow. There are also some inconsistent indentations. There are too many parameters in Table 1 that are used in the algorithm. Also, why are you not using a lower index?

The next problem of the paper is its limited evaluation:

  1. The geographical area is always rectangular. This is not the case in real-life deployments.
  2. The base-station is always in the middle. This is also unrealistic in many scenarios. Base-station is very often placed at the edge of the WSN.
  3. There is no mention of what simulator was used whatsoever. No specification of the HW, no running time.
  4. It seems that only one network was generated for each area. For each network type, there should be at least 10 different networks generated. Additionally, several different runs on each network should be executed and the results should show the averages with standard deviation.
  5. What is the importance of the grid? Who chooses the grid? How important is it to choose the right size of the grid?
  6. The results are described in a very limited manner. There are only some absolute numbers shown, but not an improvement in percentage. For example, it would be much better to write that the PSOGA extends the lifetime of the network by X% or increase the throught by X%.
  7. More importantly, explain why your algorithm performs better.
  8. What is the computational overhead of PSO vs PSOGA and more imporantly vs LEACH?
  9. How often the routes need to be recomputed? how long does it take?
  10. What happens if a CH dies? who takes after him? 
  11. In case study #4 the difference between "with" and "without" PSOGA seems to be very limited. The same applies to case study #5.
  12. Isn't "without" PSOGA just "PSO"?

Minor corrections:

Figure 4a has axis 0-200 while in the description of the experiment you claim the area is 100x100. Also, the "(a)" figures (4a, 5a, 6a, 7a) are not consistent.

Figure 5d ends at epoch 2000 while all other figures end at epoch 2500.

The computation on lines 302-304 is not easy to follow. You should add the description of how this is computed.

Zig-zag path generation is not widely known and should be included in the paper.

Overall, the paper presents a very limited contribution and the evaluation of this contribution needs much improvement. Therefore, I recommend this paper for a major revision.

Author Response

All the comments made by Reviewer is addressed in word file

Reviewer 2 Report

The work, when compared to the literature, has a low scientific contribution. Therefore, We suggest that authors increase the number of related works analyzed. Additionally, the authors could demonstrate their contribution in more detail.

The authors do not explain the time needed for the algorithm to choose the most efficient routes. Thus, it is unclear whether the protocol allows dynamic picking routes among nodes during the routing process.

Another point that must be analyzed is the interference impact between the network nodes. For this, simulation software such as NS3 could be used. Another possibility would be implementing it in a real scenario. In that case, the authors could use a smaller scale scenario than the proposed scenario.

The authors should consider the possibility of node failures and the impact on energy consumption. For example, WSN networks have a probability of failure of their nodes. Consequently, in some cases, selecting the best route should consider the failure of the node.

Author Response

(The authors gave the same response as above.)

Round 2

Reviewer 1 Report

The text has been significantly improved, however, there are still many issues.

  1. The new manuscript contains a lot of differently coloured text and it is not explained which colour means what.
  2. Some sections with lots of crossed and added text are extremely hard to read.
  3. Making a latexdiff is not a proper way how to highlight changes to the text.
  4. In the response to the reviewer graphs from 10 runs of experiments are shown, but in separate figures which makes it impossible to compare.
  5. Figure 5 does not make any sense. How can you plot just the standard deviation? What kind of information should it give to the reader? Standard deviation is always shown with the value and shows by how much this value can vary.
  6. Line 323: why is it $Pinitial$ and not $P_{initial}$? Same applies to $Pmax$.

Because of the difficulty to read and navigate in the paper, I did not fully read the paper and I am not able to fully comment on whether all the changes are sufficient for the publication of the paper.

Reviewer 2 Report

The authors improve the work concerning part of my previous questions. However, I cannot see a significant contribution so far for acceptance.

The authors consider outside the scope of the work the interference between nodes. However, interference between nodes significantly impacts the battery lifetime of nodes. Thus, I believe the authors could review the position of not considering the interference between the nodes.

The answer to the question about the possibility of node failures and the impact on energy consumption was not clear. Additionally, it is well known that nodes in WSN networks have a probability of failure. 

Just reporting that when one cluster head fails, another takes its place does not explain the impact of possible failures on more efficient path choices. Therefore, please take into account the probability of failure of nodes in the proposed solution.
